# Ketogenic Diets in the Management of Lennox-Gastaut Syndrome—Review of Literature

**DOI:** 10.3390/nu14234977

**Published:** 2022-11-23

**Authors:** Urszula Skrobas, Piotr Duda, Łukasz Bryliński, Paulina Drożak, Magdalena Pelczar, Konrad Rejdak

**Affiliations:** Department of Neurology, Medical University of Lublin, 20-090 Lublin, Poland

**Keywords:** epilepsy, epileptic syndromes, Lennox-Gastaut syndrome, diet therapy, ketogenic diets

## Abstract

Epilepsy is an important medical problem with approximately 50 million patients globally. No more than 70% of epileptic patients will achieve seizure control after antiepileptic drugs, and several epileptic syndromes, including Lennox-Gastaut syndrome (LGS), are predisposed to more frequent pharmacoresistance. Ketogenic dietary therapies (KDTs) are a form of non-pharmacological treatments used in attempts to provide seizure control for LGS patients who experience pharmacoresistance. Our review aimed to evaluate the efficacy and practicalities concerning the use of KDTs in LGS. In general, KDTs are diets rich in fat and low in carbohydrates that put the organism into the state of ketosis. A classic ketogenic diet (cKD) is the best-evaluated KDT, while alternative KDTs, such as the medium-chain triglyceride diet (MCT), modified Atkins diet (MAD), and low glycemic index treatment (LGIT) present several advantages due to their better tolerability and easier administration. The literature reports regarding LGS suggest that KDTs can provide ≥50% seizure reduction and seizure-free status in a considerable percentage of the patients. The most commonly reported adverse effects are constipation, diarrhea, and vomiting, while severe adverse effects such as nephrolithiasis or osteopenia are rarely reported. The literature review suggests that KDTs can be applied safely and are effective in LGS treatment.

## 1. Introduction

Epilepsy is a common neurological disease with approximately 50 million patients globally. Only fewer than 70% of epileptic patients can achieve seizure control with anti-epileptic drugs [1]. There are certain syndromes in which that percentage is significantly lower, e.g., Lennox-Gastaut syndrome (LGS). LGS is a rare, severe, early-onset developmental epileptic encephalopathy characterized by a triad of multiple drug-resistant seizure types, a specific electroencephalography pattern showing bursts of slow spike-wave complexes or generalized paroxysmal fast activity, and intellectual impairment. Not all patients have all of the core seizure types (e.g., tonic, atonic, and atypical absences), especially at onset [2]. The tonic type of seizure is seen in all patients with LGS, but may not be present at the time of its onset [3]. The incidence of LGS is estimated at 0.1 to 0.28 per 100,000 people per year; the lifetime prevalence at the age of ten years amounts to 0.26 per 1000 children [4]. It is estimated that LGS patients account for 1–10% of cases of childhood epilepsy and 1–2% of all epilepsy patients [2]. There are identifiable and non-identifiable causes of LGS. Approximately, 65 to 75% of patients have an identifiable cause. These include brain damage (e.g., head injury), perinatal complications (e.g., birth asphyxia, intrauterine growth retardation, kernicterus), congenital central nervous system malformations (e.g., tuberous sclerosis), infections (e.g., meningitis, sepsis), or metabolic disorders [2,3]. When LGS has no apparent cause, a genetic predisposition or etiology is probable [2]. The mutation of genes involved in human brain development (e.g., the forkhead box G1 (FOXG1), or chromodomain-helicase-DNA-binding protein 2 (CHD2) genes) and the gene for presynaptic protein dynamin 1 (DNM 1) have been found to be associated with this syndrome. Valproate, lamotrigine, and topiramate are considered the first-choice drugs. According to randomized control trials, other anti-epileptics reported to be effective are clobazam, felbamate, and rufinamide. It is important to start with as few drugs at a time as possible, in the lowest possible doses. If the first drug fails, change to another drug, but if the second drug is also not effective, a second agent can be added to the existing regimen [3]. As available pharmacologic treatments for seizures are limited in their efficacy, several non-pharmacological methods are used, including surgical and dietary interventions. Surgical procedures can lead to control in a select subset of patients suitable for that kind of treatment but still, a significant number of patients are left with uncontrolled seizures. The ketogenic diet and related diets have proven to be useful in drug-resistant childhood epilepsy [5]. The main purpose of this article is to evaluate the effectiveness and practicalities of using ketogenic dietary therapies (KDTs).

## 2. Search Strategy

A literature search was performed using PubMed keywords: “ketogenic diet”, “medium chain triglyceride diet”, “low glycemic index treatment”, “modified Atkins diet”, and “Lennox-Gastaut syndrome”. In this review, only publications in the English language were used. The efficacy of KDTs was determined by the percentage of patients that achieved at least 50% seizure reduction and seizure-free status.

## 3. Ketogenic Dietary Therapies

KDTs are well-established, nonpharmacological treatments used for children and adults with pharmacoresistant epilepsy [6]. Their mechanisms of action rely on putting the organism into a ketogenic state, similar to starvation, which has been reported as a treatment for epilepsy since ancient times [7]. KDT refers to any diet therapy in which dietary composition results in a ketogenic state of human metabolism. The diet generally refers to a high-fat, low-carbohydrate, and moderate-protein diet. Following the development of the classic ketogenic diet (cKD), new diets have been proposed in an attempt to increase retention and savoriness while imitating the effects produced by the original diet. Currently, there are four major KDTs—cKD, the modified Atkins diet (MAD), the medium-chain triglyceride diet (MCT), and the low glycemic index treatment (LGIT). The compositions of the four main KDTs differ (Table 1) and the data comparing the efficacy of the above-mentioned diets are limited [8].

The cKD and the MCT diet have been in existence the longest and are typically started in the hospital by a dietitian and neurologist. cKD is the oldest and the most researched diet among dietary therapies for epilepsy. It was designed in the year 1923 by doctor Russell Wilder from the Mayo Clinic and was designed to particularly treat this disease [9,10]. In the cKD, the fat source is largely long-chain triglycerides (LCT), which are gained mainly from standard foods. cKD is typically administered in a 4:1 ratio of fat to carbohydrate and protein, providing 90% of the total calories from the fat. MCT oils provide more ketones per kilocalorie of energy than LCTs. This enlarged ketogenic potential means less total fat is needed in the MCT diet, which allows the inclusion of more carbohydrates and protein and more potential food choices [6]. In both of the above-mentioned KDTs, calculations and arrangements concerning the diet and education of a patient and their family should be conducted by a dietitian [6,11].

As the cKD and MCT can be too restrictive, alternative diets were created. They are more liberal versions of the cKD, which are less restrictive on protein and calorie intake, more affordable for patients, and can be similarly effective in the treatment of epilepsy. The MAD was created at Johns Hopkins Hospital in 2003, primarily for children with behavioral difficulties and adolescents whose parents and neurologists were unwilling to start on the cKD [12]. The MAD is a high-fat, low-carbohydrate therapy which typically provides approximately a 1:1–1.5:1 ketogenic ratio, but no set ratio is required and some children can achieve a ratio as high as 4:1. The initial daily carbohydrate consumption on the MAD is approximately 10–15 g with a possible increase to 20 g per day after 1–3 months. In addition, there is no restriction on protein, fluids, or calories, making meal planning easier. Detailed calculations are not required [6].

The LGIT was introduced as an elasticized version of the cKD and was first found to be successful in Massachusetts General Hospital in 2005 [13]. The diet is more liberal and allows 40 to 60 g of carbohydrates per day, but restricts sources of carbohydrates to a glycemic index of less than 50 to prevent postprandial increases in blood glucose. Fats and proteins are unrestricted. This diet is preferred by adolescents because of their difficulty in following the cKD. Admission to hospital is not required to implement the treatment [8].

**Table 1 nutrients-14-04977-t001:** Examples of the major ketogenic diet types used in certain reviewed studies.

Ketogenic Diet Types	Components of the Major Ketogenic Diets			References
	Fat	Carbohydrate	Protein	
Classic ketogenic diet (cKD) 4:1	4 g of fat to 1 g of protein plus carbohydrate			Zhang Y et al., 2016, [14]
Modified Atkins diet (MAD)	not restricted, actively encouraged	restricted to 10 g/d	not restricted	Sharma S et al., 2015, [15]
Medium-chain triglyceride diet (MCT)	4.3 g/kg/d, of which 70% were MCT	16% of total calories		Rosenthal E et al., 1990, [16]
Low glycemic index treatment (LGIT)	60% of total calories	10% of total calories	30% of total calories	Kim SH et al., 2017, [17]

## 4. Mechanisms of KDTs’ Antiseizure Effect

Despite the antiseizure effect of ketosis being known from ancient times [7], the mechanisms of KDTs’ action have not been fully elucidated [18]. Among proposed theories, the abundance of ketone bodies (KBs) and fatty acids, the decreased glucose supply, and the impact on gut microbiota are mainly considered, inducing the modifications of neuronal metabolism, neurotransmitters, and ion channels [19,20,21]. Many authors suggest that the antiseizure effect of KDTs relies on a combination of several mechanisms [20,22,23,24]. 

KBs and fatty acids can generate direct changes in organism homeostasis. Acetoacetate [25] and beta-hydroxybutyrate [25,26] were shown to induce activation of ATP-sensitive potassium (KATP) channels [25,26], while polyunsaturated fatty acids (PUFA) and, presumably, KBs activate two-pore domain potassium (K2P) channels [27], increasing the seizure threshold via neuronal hyperpolarization [25,26,27]. A medium-chain fatty acid—decanoic acids—was shown to independently contribute to seizure control through alpha-amino-3-hydroxy-5-methyl-4-isoxazolepropionic acid (AMPA) receptor inhibition [23]. On the other hand, one of the first KBs-related hypotheses (the pH hypothesis) is no longer considered convincing, as some studies showed that KDTs do not make the blood nor brain tissue more acidic [28].

KDTs conditions also promote neuronal stability via their effects on the neurotransmitters mainly affecting gamma-aminobutyric acid (GABA) [19]. Upregulation of glutamic acid decarboxylase [29] and reduction in GABA transaminase activity [30] were suggested to result in enhanced GABA concentration. Regarding glutamate, the impact of KDTs on its levels is inconstant, and further research in this field is needed. Norepinephrine, dopamine, serotonin, galanine, and neuropeptide Y are highlighted among other neurotransmitters presumably involved in the antiseizure effect of KDTs [19].

Regarding energy metabolism, prolonged KDT implementation promotes mitochondria synthesis and concentration, increasing adenosine triphosphate (ATP) generation and enhancing the brain’s resistance to metabolic stress [19,28]. Reduced glycolysis presumably contributes to the antiseizure effect, as an intravenous glucose bolus reverses ketonemia and the antiseizure effects of KDTs rapidly [31]. Additionally, glucose restriction in KDTs promotes the pentose–phosphate pathway and is suspected to reduce reactive oxygen species (ROS)-dependent cell damage owing to enhanced nicotinamide adenine dinucleotide phosphate (NADPH) production [32]. 

Of late, the attention of researchers has focused on the KDTs’ impact on the microbiome–gut–brain axis. In an innovative study on two mice models, Olson et al. [33] showed that changes in gut microbiota were necessary for the antiseizure effects of KDTs, and the effect was mediated via elevating GABA/glutamate ratio in the hippocampus. Regarding humans, microbiome modification via probiotics was shown to promote seizure reduction [34], and several studies indicated that KDTs can alter the gut microbiome [21,35,36]. Some authors suggest that in the future, KDTs could even be replaced with prebiotic and probiotic supplementation or fecal microbial transplants (FMTs) [21].

## 5. Efficacy of cKD

The efficacy of cKD in the treatment of LGS is well-documented in scientific research. A literature review conducted by Lemmon ME et al. [37], which included 18 papers from the years 1989–2010 concerning applying cKD in, overall, 189 children with LGS, showed that after 3–36 months of using this diet, 47% of investigated children experienced ≥50% seizure reduction, whereas 16% of children achieved seizure-free status. Moreover, it was proven that the cKD causes a reduction in multiple types of epileptic seizures, including tonic, atonic, or atypical absence seizures. Additionally, a study conducted by Lemmon ME et al. showed that among 71 children who took part in the study, 32 (45%) were able to reduce the dose of parallelly taken anticonvulsant drugs while using cKD [37]. Reviewed papers from the years 2012–2020 show that a ≥50% seizure reduction was observed in 17–73% of patients, whereas a complete seizure reduction was observed in 1–23.5% of patients [6,14,37,38,39,40,41]. 

No predictive factors were found for the reduction in the number of seizures after using cKD in LGS. Research by Lemmon ME et al. [37] showed that age, gender, presence of side effects, etiology of LGS, number of previously taken anticonvulsants, and history of seizures during infancy were not predictive of >90% seizure reduction after 12 months of using cKD [37]. The reviewed data regarding the efficacy of cKD in the treatment of LGS is presented in Table 2.

**Table 2 nutrients-14-04977-t002:** Efficacy of classic ketogenic diet in Lennox-Gastaut syndrome.

Number of LGS Patients	Time of Assessment	Effectiveness (≥50% Seizure Reduction)	Effectiveness (Seizure-Free Status)	Side Effects	Additional Information	References
71	3rd month 6th month 12th month	70% 49% 42%	6% 1% 1%	Constipation, weight loss, kidney stones, hyperlipidemia, poor linear growth	-	Lemmon ME et al., 2012, [37]
15	18th month	33% *	20% *	Vomiting, hypoglycemia	5 of 20 children (25%) discontinued the diet within the first year due to a lack of effectiveness	Caraballo RH et al., 2014, [38]
68	Mean follow-up duration of 19.3 years	N/A	24%	N/A	-	Kim HJ et al., 2015, [39]
47	1st month 3rd month 6th month	32% 49% 43% *	4% 4% 10% *	Hyperlipidemia, gastrointestinal problems, fatigue, drowsiness	7 out of 47 patients discontinued the diet due to lack of effectiveness or inability to adhere to the diet	Zhang Y et al., 2016, [14]
7	12th week	29%	14%	Digestive symptoms, hypoglycemia	-	Wu Q et al., 2018, [40]
18	1 year 2 years	27.8% * 27.8% *	11.1% * 11.1% *	Poor oral intake, metabolic acidosis, osteopenia, gastrointestinal problems, bleeding	The study involved patients with mitochondrial dysfunction. 16 patients were prescribed a KD with a lipid: nonlipid ratio of 4:1. Two patients were prescribed a KD with a lipid: nonlipid ratio of 3:1; the remaining two were prescribed the MAD (see Table 3)	Na JH et al., 2020, [41]
1	3rd month 6th month	100% 100%	0% 0%	The patient presented no side effects.	The study is a case report of a pediatric patients with tyrosinemia type 1.	De Lucia et al., 2015, [42]

Abbreviations: LGS—Lennox-Gastaut syndrome; KD—ketogenic diet; MAD—modified Atkins diet; N/A—not available. *—quantity was calculated based on data from the publication.

## 6. Efficacy of Alternative Ketogenic Diets

Although alternative ketogenic diets were introduced to provide better tolerability and palatability than cKD, their efficacy is the most important issue. The comparison of the efficacy of MAD and cKD was assessed among the general population of children and adolescents with epilepsy. In a meta-analysis conducted by Razaei et al. [43], no statistically significant differences in efficacy were reported between MAD and cKD. Regarding LGS patients, a study conducted by Sharma S et al. [15] comprised of 25 pediatric LGS patients treated with MAD. After three months, nearly half of the patients had a ≥50% reduction in seizure frequency. After six months, 14 children discontinued the diet; however, all of the children who remained on the diet had more than a 50% reduction in seizure frequency both at the 6th and 12th month of the study [15]. In another retrospective study, Na JH et al. [41] assessed the efficacy of several KDTs in LGS patients, including two LGS patients who were prescribed MAD. One of the patients achieved 25% seizure reduction, while the second patient did not show any seizure reduction (Table 3) [41]. 

In another study by Kim et al. [17], LGIT was used as a treatment option on 36 epileptic patients. Only 12 patients had LGS and the rest of the patients had different epileptic syndromes. A positive response to prescribed treatment was seen in 75% (*n* = 9) of the LGS patients. The authors claimed that LGIT would be particularly effective among patients who previously responded favorably to cKD and could be a promising less-restrictive alternative after the success of cKD [17]. According to a case report by Kumada T et al. [44], LGIT allows for reducing the frequency of tonic seizures and myoclonic seizures. This case report described treatment using Japanese ethnic foods for a 13-year-old girl with LGS. At the age of 11 years, the patient was treated with MAD, but after two weeks she refused the diet. At the age of 13 years, LGIT was applied. After one year of therapy, the efficacy of the diet has been sustained (Table 3) [44].

According to a case report by Rosenthal E et al. [16], MCT served in parenteral nutrition provided the control of seizures and allowed normal daily functioning of one LGS patient. This case study shows that intravenous (iv) MCT can be used as a short-term method of maintaining ketosis for seizure control, but this method may be linked to complications (Table 3) [16].

**Table 3 nutrients-14-04977-t003:** Efficacy of alternative ketogenic diets in Lennox-Gastaut syndrome.

Number of LGS Patients	Ketogenic Diet Types	Time of Assessment	Effectiveness (≥50% Seizure Reduction)	Effectiveness (Seizure-Free)	Side Effects	Additional Information	References
25 11 9	MAD	3rd month 6th month 12th month	48% 44% * 36% *	8% * 12% * 12% *	Constipation, vomiting, anorexia;	After a year 64% (*n* = 16) of the patients discontinued the KDT.	Sharma S et al., 2014, [15]
2	MAD	3rd month 6th month 9th month 12th month 24th month	0% * 0% * 0% * 0% * 0% *	0% * 0% * 0% * 0% * 0% *	Osteopenia	One patient in the 3rd month of the MAD use had a 25% reduction rate of seizures.	Na JH et al., 2020, [41]
1	LGIT	12 months	N/A	N/A	N/A	After 1 month, the tonic seizures during sleep had decreased to a frequency of once or twice per month. After 1 month, the myoclonic seizures during the awake state had disappeared.	Kumada T et al., 2013, [44]
12	LGIT	3rd month 6th month 12th month	N/A N/A N/A	N/A N/A N/A	Transient diarrhea, laboratory abnormalities–reduced serum total carbon dioxide, hypercholesterolemia, increased alanine aminotransferase, increased lipase increased blood urea nitrogen;	75% of patients with LGS had a positive response to LGIT.	Kim, SH et al., 2017, [17]
1	parenteral MCT	6 months	N/A	N/A	Abnormal liver function tests and severe iron deficiency anemia, increased levels of serum triglyceride and cholesterol	Intravenous nutrition was applied because oral feeding and nasogastric and nasoduodenal feeding were precluded.	Rosenthal E et al., 1990, [16]

Abbreviations: LGS—Lennox-Gastaut syndrome; KDT—ketogenic dietary therapy MAD—modified Atkins diet; N/A—not available; LGIT—low glycemic index treatment; MCT—medium-chain triglyceride diet. *—quantity was calculated based on data from the publication.

## 7. Safety and Tolerability of Ketogenic Dietary Therapies

The literature reports from the general epileptic population show that KDTs are generally considered as safe treatment options [40,41,45]. They are applied in adults, as well as in children [46], showing good tolerability for patients as young as six weeks [47]. Additionally, cKD may be used in groups of LGS patients presenting certain metabolic abnormalities. A study conducted by Na et al. [41] suggests that KDTs can be used to treat LGS in patients with mitochondrial dysfunction, while De Lucia et al. [42] suggested that cKD with restricted phenylalanine and tyrosine can be used in patients with tyrosinemia type 1. In addition, van der Louw et al. [48] showed the case series of two pregnant women with epilepsy. One was treated with MCT as a monotherapy, while the second was treated with MAD and lamotrigine. Both women achieved seizure reduction, and, after labor, their children presented normal growth and development at a follow-up of 12 and 8 months, respectively. On the other hand, the child of the second woman presented bilateral ear deformities of unknown significance. The authors claimed that KDTs may be effective during pregnancy, but their safety has to be established [48]. Common adverse effects of KDs are vomiting, constipation, and diarrhea [40,49]. However, in the meta-analysis of randomized controlled trials, the above-mentioned adverse effects were also commonly reported by the patients from the usual-care group with diets having no known effects on epilepsy [49]. 

In the case of LGS, reviewed research papers show that the use of cKD could have certain side effects including gastrointestinal disturbances, such as diarrhea, constipation or vomiting, hypocalcemia, kidney stones, hyperuricemia, and metabolic acidosis [50]. Patients subjected to cKD also experience problems with the cardiovascular system, kidneys, and skeletal system [51]. Moreover, research shows that this diet could cause growth retardation in children [52,53]; however, the data regarding growth retardation is mixed [6]. It was proven that patients who experience side effects have a higher risk of non-adherence to this diet [49].

Compared to cKD, a lower occurrence of side effects is an advantage of LGIT. As a result, this diet is reported as easier to use and continue [19,44]. The side effect among all patients in one of the reported studies was transient diarrhea. In addition, laboratory abnormalities were observed—reduced serum total carbon dioxide, hypercholesterolemia, increased alanine aminotransferase, increased lipase, and increased blood urea nitrogen. However, mentioned conditions did not require additional management or medication [17].

Similarly, MCT, which enables higher carbohydrate and protein, seems to be more favorable and palatable for children than the cKD [5,19]. However, the authors of the case study concerning iv MCT stated that this method had complications—abnormal liver function tests and severe iron deficiency anemia of unknown etiology. In addition, serum triglyceride and cholesterol levels increased, but they decreased after a reduction in lipid infusion and the use of an antihyperlipidemic drug [16].

In the reviewed studies concerning MAD, patients reported mild side effects including constipation, vomiting, and anorexia. In addition, none of the children from that study had serious side effects, such as deranged renal or hepatic functions [15]. On the other hand, the authors of the second MAD publication reported that one of the patients showed osteopenia as a side effect of the diet [41]. However, MAD is better tolerated by patients than the cKD and they are more willing to follow MAD (56%) than cKD (38%) [54]. MAD also presents several advantages regarding patient comfort and benefits for the healthcare system. There is no need for hospitalization during dietary treatment onset. Lower costs of the diet and better tolerability suggest that MAD may be a promising substitute for patients who do not tolerate cKD well. These aspects are even more significant in some regions of the world, where no KD centers are available [43].

## 8. Problem of Attrition

A high attrition rate of patients receiving KDTs is still a major concern among the general epileptic population [43,49]. The most frequently reported reasons for cKD discontinuation are a lack of expected efficacy [14,49,55], dietary intolerance and strictness [20,55], and its low palatability [17,49]. However, the authors of the meta-analysis of randomized controlled trials regarding KDTs reported no statistically significant differences in attrition rates between the KDs group and the usual care group, with attrition rates of 8–38% and 2.5–32%, respectively [49]. Similarly to reports from general epileptic patients, the most frequent discontinuation reason in reviewed publications was a lack of effectiveness, both for the cKD [56] and MAD [15].

## 9. Conclusions

According to the authors of the above-mentioned studies, KDTs are effective in LGS treatment. KDTs are reported as safe treatment options and can be administered to different populations of epileptic patients. The most common complications include constipation, diarrhea, and vomiting, while severe adverse effects are rarely reported. Several less-restrictive KDTs, including MCT, MAD, and LGIT, are worth considering, due to their better tolerability, lower treatment costs, and easier administration.

## 10. Future Directions

As many of the studies comprise small patient samples, further large randomized controlled trials would be needed to clarify the issue of KDTs efficacy. What is more, future studies should address the issues of patients’ quality of life and cognitive outcomes more frequently, and more consistency in reporting seizure reduction rates (e.g., based on ≥50% seizure reduction rates, and seizure-free status rates) would be beneficial. The paucity of studies concerning adults and infants indicates the need to evaluate these patient populations, particularly.

## Data Availability

Not applicable.

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
