# Peer review of "Ketogenic Diets in the Management of Lennox-Gastaut Syndrome—Review of Literature"

_nutrients, 2022, doi:10.3390/nu14234977_

Round 1

Reviewer 1 Report

Skrobas et al., in this review have detailed the use of ketogenic diet as a treatment for epilepsy with a special focus on Lennox-Gastuat syndrome.

I have the following suggestions for the review.

1] A section explain the potential molecular mechanism of the ketogenic diet on epilepsy will be a good section to include.

2] Use original primary references in Line 82 and line 88.

Author Response

Dear Sir or Madam,

Thank You very much for the review of the manuscript entitled: “Ketogenic Diets in the Management of Lennox-Gastaut Syndrome–Review of Literature”.

Manuscript changes are highlighted in yellow.

Comment 1

A section explain the potential molecular mechanism of the ketogenic diet on epilepsy will be a good section to include.

Revision and my comment

We included an adequate section, providing a brief explanation of the Ketogenic Dietary Treatments’ mechanisms. We also added abbreviations of the words that appeared in the new section.

Comment 2

Use original primary references in Line 82 and line 88.

Revision and my comment

We have changed the reference in line 82 and added the original publication in line 88 (now lines 98 and 105, respectively). At present, the original primary bibliography is used.

We do honestly hope that it will satisfy You and improve the quality of our work. Once again, we are very grateful for Your review and remain open if You have any other remarks or suggestions that will make our work merit publication in “Nutrients”.

Yours faithfully

Piotr Duda on behalf of the Authors

Reviewer 2 Report

In a review paper, the authors present data regarding the effectiveness of various types of ketogenic diets used in the treatment of  Lennox-Gastaut Syndrome.

Minor comments:

Does the abstract have to end with such an unoptimistic sentence?

I would suggest expanding the introduction with more data of the Lennox-Gastaut Syndrome (causes, mechanisms, characteristic symptoms, types of seizures, pharmacotherapy), also a little more information on the mechanism of the ketogenic diet would be good to include.

I think it is necessary to organize the information contained in the tables:

Tab. 1, „References” instead „Authors, year, and reference numer”,  it will be better to move to the last column (Tab. 2 and 3)

„Ketogenic Diet Types” Instead „Ketogenic Diet Therapies protocol” (Tab. 2)

Maybe it is enough to give the effectiveness in percentages, I also suggest reducing the text in the tables, especially in Table 3 in the „additional information” column.

Author Response

Dear Sir or Madam, thank You very much for the review of the manuscript entitled: “Ketogenic Diets in the Management of Lennox-Gastaut Syndrome–Review of Literature”.

Manuscript changes are highlighted in yellow.

Comment 1

Does the abstract have to end with such an unoptimistic sentence?

Revision and my comment

We removed a sentence regarding the reasons for attrition from the abstract and added a sentence referring to the conclusions instead.

Comment 2

I would suggest expanding the introduction with more data of the Lennox-Gastaut Syndrome (causes, mechanisms, characteristic symptoms, types of seizures, pharmacotherapy), also a little more information on the mechanism of the ketogenic diet would be good to include.

Revision and my comment

We expanded the introduction with more data as it was suggested. We also included a section “Mechanisms of KDTs’ antiseizure effect”, providing a brief explanation of the KDTs’ mechanisms of action. Adequate abbreviations of the words that appeared in the new paragraph were added.

Comment 3

I think it is necessary to organize the information contained in the tables:

Tab. 1, „References” instead „Authors, year, and reference numer”,  it will be better to move to the last column (Tab. 2 and 3)

„Ketogenic Diet Types” Instead „Ketogenic Diet Therapies protocol” (Tab. 2)

Maybe it is enough to give the effectiveness in percentages, I also suggest reducing the text in the tables, especially in Table 3 in the „additional information” column.

Revision and my comment

In Table 1., we changed the column’s heading "Authors, year, and reference number" to “References”. We moved the column “references” to the last column in all of the tables. We also provided the data about effectiveness in percentages only. Furthermore, we replaced „Ketogenic Diet Therapies protocol” with „Ketogenic Diet Types” in Table 1. and “Type of diet” with „Ketogenic Diet Types” in Table 3. We also reduced the text in Table 3.  in the „additional information” section.

We do honestly hope that it will satisfy You and improve the quality of our work. Once again we are very grateful for Your review and remain open if You have any other remarks or suggestions that will make our work merit publication in “Nutrients”.

Yours faithfully

Piotr Duda on behalf of the Authors

Round 2

Reviewer 1 Report

No further corrections to be made.